# VIRT: Vision Instructed Transformer for Robotic Manipulation

## Abstract

Robotic manipulation, owing to its multi-modal nature, often faces significant training ambiguity, necessitating explicit instructions to clearly delineate the manipulation details in tasks. In this work, we highlight that vision instruction is naturally more comprehensible to recent robotic policies than the commonly adopted text instruction, as these policies are born with some vision understanding ability like human infants. Building on this premise and drawing inspiration from cognitive science, we introduce the robotic imagery paradigm, which realizes large-scale robotic data pre-training without text annotations. Additionally, we propose the robotic gaze strategy that emulates the human eye gaze mechanism, thereby guiding subsequent actions and focusing the attention of the policy on the manipulated object. Leveraging these innovations, we develop VIRT, a fully Transformer-based policy. We design comprehensive tasks using both a physical robot and simulated environments to assess the efficacy of VIRT. The results indicate that VIRT can complete very competitive tasks like "opening the lid of a tightly sealed bottle", and the proposed techniques boost the success rates of the baseline policy on diverse challenging tasks from nearly 0% to more than 65%.

## 1 Introduction

*"Seeing comes before words."* – John Berger

In robotic manipulation, a policy is trained to manipulate objects according to environment observations and task requirements Billard & Kragic (2019). The key insight that supports this work is **existing robotic policies are akin to human infants**, who are born with visual perception and reasoning abilities but do not comprehend natural language according to previous cognitive science literature Colombo & Mitchell (2009). Specifically, visual signal serves as the primary information source of recent robotic policies, and the backbones of these policies are pre-trained with large-scale image datasets before the robotic data based training He et al. (2016); Oquab et al. (2024). Therefore, the policies begin with a basic visual understanding capability like human infants. By contrast, natural language inputs are rarely incorporated into the process of pre-training these backbones, suggesting the lack of natural language knowledge in these policies. Moreover, bridging this knowledge gap demands extensive image-text alignment training Radford et al. (2021), which is typically impractical for developing real-time robotic policies.

Given these thoughts, we can deduce that the currently common practice of utilizing natural language to instruct policies about task requirements (*e.g.*, what task to do or which object to manipulate) is unsuitable Zitkovich et al. (2023); Kim et al. (2024b). To address this problem, we introduce the concept of vision instruction, which involves using images to guide policies. In this work, we concern the two main training settings in robotic manipulation learning, *i.e.*, task-unspecified pre-training Mees et al. (2024) and task-specified training Kim et al. (2024a). The optimization objective of the former one is to derive a policy with general manipulation knowledge, which can then be used as the initial model for downstream tasks. Differently, the latter setting focuses on creating a policy model that can effectively control a robot to complete specific tasks. This work focuses on exploring how vision instruction can be leveraged to enhance these two training settings through replacing the original inappropriate text instructions.

Existing task-unspecified pre-training methodologies predominantly utilize task-aware training data, which means the task content of each manipulation trajectory is known and often represented as

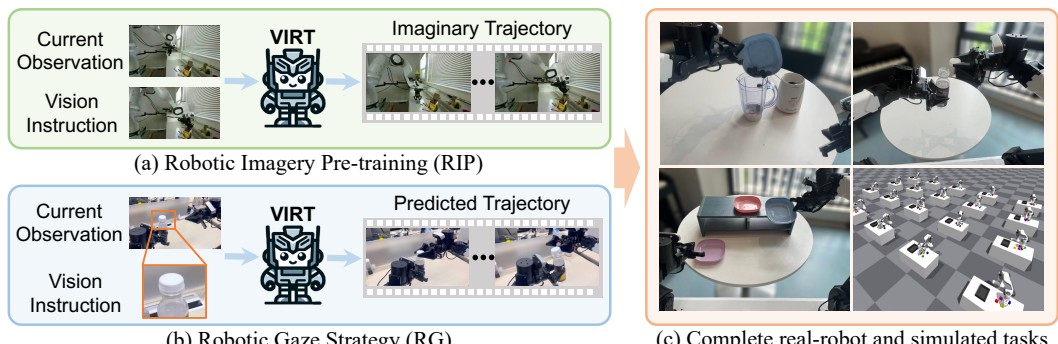

Figure 1: In this work, we first adopt the proposed robotic imagery pre-training paradigm to pre-train VIRT based on large-scale robotic manipulation data. Then, we fine-tune the pre-trained policy on specific downstream tasks with the robotic gaze strategy. After these two phases of training, VIRT is able to complete diverse challenging tasks in both real-robot and simulated environments.

language based task descriptions O'Neill et al. (2024). Nonetheless, as discussed before, without sufficient image-text pair pre-training, policy networks fail to really understand these descriptions. Besides, some literature indicates that a large volume of network parameters is crucial for language understanding Achiam et al. (2023); Liu et al. (2024), whereas robotic policies often have limited parameters due to real-time deployment constraints. Last but not least, not all data is labeled with task descriptions, which hinders the scaling up of pre-training. To handle these limitations, we design a novel pre-training paradigm termed robotic imagery pre-training (RIP). This approach is inspired by the motor imagery mechanism in cognitive science, which demonstrates that imagining actions in human brains without execution shares similar cortex with actually performing these actions, and just practicing these actions in mind results in significant positive performance transfer to real operations Decety (1996). Inspired by this mechanism, our proposed RIP utilizes visual observations from the initial and final frames of manipulation trajectories to train a policy network to mentally imagine the entire trajectory. This process, depicted in Fig. 1 (a), eliminates the need for task descriptions and facilitates the scalability of pre-training across diverse data sources.

In task-specified training, a key challenge is the ambiguity in manipulation trajectories. For instance, in a table-clearing task with multiple objects, the policy must determine which object to grasp first. To handle this problem, vision instruction should assist in scheduling manipulation procedures. Drawing inspiration from the gaze anchoring hypothesis Land & Hayhoe (2001) in cognitive science, we propose a robotic gaze (RG) strategy, as illustrated in Fig. 1 (b). Specifically, the gaze anchoring hypothesis believes that eye gaze is seeking out objects for future use and setting up the operations to perform. To replicate this mechanism, we employ a lightweight detector Wang et al. (2024) to recognize the object to manipulate at each action step. Afterwards, we crop out the recognized object region from the image and enlarge it. The enlarged image region is fed to the policy, guiding it in object manipulation. Notably, the enlarging image operation is non-trivial, because robotic manipulation requires clear perception of object details. However, enhancing the resolution of the entire image incurs a high computational cost. By contrast, resizing only the target object region is much more efficient. This design of enlarging the concerned region in an image mirrors human vision, where the eye perceives a small, focused area with sharp clarity while the surrounding regions remain blurred Stewart et al. (2020).

By integrating the aforementioned techniques, we develop a fully Transformer-based policy, namely **V**ision **I**nstructed **R**obotic **T**ransformer (VIRT) 🤖. VIRT utilizes image observations, historical trajectories, and vision instructions to control a robot for dexterous manipulations with a rapid response speed. We evaluate VIRT based on extensive real-robot and simulated tasks. For real-robot experiments, we devise a task of pouring blueberries into a juicer cup to verify the multi-object manipulation ability, a task of opening the lid of a tightly sealed bottle to validate the bimanual dexterous manipulation precision, and a task of cleaning the plates on a table according to the online instructed order to evaluate the instruction following capability. In simulated experiments, we build a real-time human hand pose acquisition system to teleoperate robotic hands in Isaac Gym Makoviychuk et al. (2021) and design several tasks of transporting and stacking color blocks following task instructions to further analyze the characteristics of VIRT. All the experiments demonstrate the strong effec-

tiveness of the proposed techniques, indicating superior performance compared to recent popular methods like ACT Fu et al. (2024b) and Diffusion Policy Chi et al. (2023).

## 2 RELATED WORK

### 2.1 DEMONSTRATION LEARNING BASED ROBOTIC MANIPULATION

Robotic manipulation, defined as the capacity for robots to interact with and modify their surroundings, has advanced markedly due to the integration of machine learning techniques Fang et al. (2019). Among existing methodologies, demonstration learning (also known as imitation learning or behavior cloning) has garnered significant attention for its training efficiency Zhao et al. (2023). Demonstration learning enables robots to acquire complex manipulation skills by observing human demonstrations, thereby bypassing the need for explicit programming of every action. The fundamental premise of demonstration learning is that a human teacher performs a task while the robot records the relevant data, such as sensory inputs, actions, and corresponding outcomes. This recorded data is subsequently used to train models that allow the robot to replicate the demonstrated behavior in similar situations Florence et al. (2022).

After continuous efforts paid by the research community, many advanced demonstration learning based policies have been developed, and they can be broadly categorized into two groups, *i.e.*, explicit policies Fu et al. (2024b) and implicit policies Chi et al. (2023). Among them, explicit policies directly map environment observations to actions, and the policy output is supervised with human demonstration trajectories by computing regression losses Fu et al. (2024a). In contrast, implicit policies define the distributions of actions with energy-based models, where predicting the next action is framed as identifying the manipulation trajectory with minimal energy Chi et al. (2024). This modeling approach allows for the natural representation of multi-modal distributions of manipulation trajectories, as multiple actions can simultaneously be assigned low energies. Consequently, some studies suggest implicit policies are more advantageous for robotic manipulation learning Florence et al. (2022). Nevertheless, we contend that explicit policies offer faster response speeds due to their simplicity, which is crucial for robotic manipulation. In addition, the iterative decoding mechanism inherent in Transformer models is similar to the denoising process in implicit policies, and thus can also handle the multi-modal ambiguity in robotic manipulation to some extent. Hence, in this work, we develop a fully Transformer-based policy adopting the explicit prediction paradigm.

### 2.2 ROBOTIC PRE-TRAINING

Recent advancements in natural language processing and computer vision demonstrate the efficacy of first pre-training models on large-scale data and then fine-tuning them for specific downstream applications Achiam et al. (2023); Wang et al. (2023b). Drawing inspirations from these successes, the robotic learning community begins to explore pre-training paradigms to enhance robotic manipulation capabilities. The principal idea behind robotic pre-training is to first expose the robotic policy to a wide range of tasks and environments, allowing it to learn generalizable representations for diverse robotic tasks Brohan et al. (2022). Subsequently, a fine-tuning phase on specific manipulation tasks is conducted, utilizing the previously gained prior knowledge to enhance performance and efficiency Zitkovich et al. (2023).

Pre-training a policy requires a substantial amount of data Fang et al. (2020). However, robotic manipulation data is expensive to collect. To mitigate this issue, some methods generate data through simulated environments based on the traditional force closure estimation algorithms Wang et al. (2023a). However, significant discrepancies in appearance and motion dynamics between simulated and real-robot data limit the effectiveness of pre-trained policies. Some efforts employ large language models to generate grasp positions for objects in 2D images Vuong et al. (2023), but this approach is constrained by its two-dimensional output, whereas robotic manipulation occurs in a three-dimensional space. Recently, collaborative efforts among various institutions have led to the creation of large-scale datasets by merging existing data sources O'Neill et al. (2024) or collecting new data across diverse scenarios Khazatsky et al. (2024). Thanks to these datasets, a handful of promising pre-trained policies are derived Kim et al. (2024b). Nevertheless, robotic manipulation trajectories remain highly ambiguous if without appropriate task instructions as prompt. Existing pre-training algorithms predominantly use text instructions to inform the policy, which restricts the pre-training effectiveness, as previously noted.

## 2.3 Robotic Instruction

Robotic manipulation learning is intrinsically a long-sequence autoregressive problem, often involving thousands of action steps within a short duration Chen et al. (2024). Therefore, a basic challenge in manipulation lies in the ability of a policy to determine the appropriate actions based on the current observation. This problem is especially serious if the task to perform involves multi-object manipulation or there are many potential operation steps Shi et al. (2023). To alleviate this problem, instructions are demanded to guide policies with task-specific information. In previous works, the instructions are mostly represented as natural language Brohan et al. (2022); Zitkovich et al. (2023), which is difficult to understand for policies as previously discussed. Alternatively, some researchers have also explored voice instructions Shi et al. (2024), but voice information similarly poses learning difficulties. By contrast, images are more readily comprehensible for policy networks, as the commonly adopted backbones of these networks are already pre-trained on extensive image datasets He et al. (2016). This pre-existing visual understanding in robotic policies is akin to the innate vision comprehension in human infants. Despite the potential of visual instructions, their applications remain unexplored in the context of task-unspecified pre-training and task-specified training of robotic manipulation. Existing studies on visual instructions have primarily focused on goal images within game-based reinforcement learning Yuan et al. (2024) and navigation Majumdar et al. (2022). This work aims to bridge the gap in exploring visual instructions for robotic manipulation.

## 3 Method

### 3.1 Problem Formulation

Robotic manipulation learning is inherently a long-sequence auto-regressive prediction problem and can be formalized as a Markov Decision Process defined by $\mathcal{M} = (\mathcal{S}, \mathcal{A}, \mathcal{P})$. Here, $\mathcal{S}$ denotes the set of all possible environmental states, $\mathcal{A}$ represents the set of possible actions by the policy, and $\mathcal{P}$ indicates the transition probability distribution for transitioning to the next state given the current state and action. At a given timestamp $t$, with the environment in state $s_t \in \mathcal{S}$, the corresponding observation $o_t$ of a policy $\pi$ is typically a function of $s_t$, denoted as $o_t = f(s_t)$. This observation $o_t$ is then used to determine the next action with respect to $a_t \sim \pi(o_t)$, where $a_t \in \mathcal{A}$. Upon executing the action $a_t$, the environment state transitions according to $s_{t+1} \sim \mathcal{P}(s_{t+1} \mid s_t, a_t)$.

To train $\pi$, we manually collect a set of demonstration data $\mathcal{D} = \{(\hat{s}_t^i, \hat{o}_t^i, \hat{a}_t^i) \mid i \in \{1, \ldots, N\}, t \in \{0, \ldots, T_i\}\}$, where $N$ is the number of demonstrations and $T_i$ denotes the length of the $i_{\text{th}}$ trajectory. Ideally, we expect the actions predicted by the policy $\pi$ to change environment states following the dynamics observed in $\mathcal{D}$, which means that $s_{t+1}$ should be akin to $\hat{s}_{t+1}$ given similar initial states $s_t$ and $\hat{s}_t$. Nevertheless, directly computing the loss between $s_{t+1}$ and $\hat{s}_{t+1}$ is infeasible due to the non-differentiable nature of environment states. Consequently, existing methods approximate this process through computing the loss between the predicted action $a_t$ and ground truth action $\hat{a}_t$.

### 3.2 VIRT Policy

In this work, we parametrize the policy $\pi$ with the proposed fully Transformer-based model VIRT, which primarily consists of 12 encoders and 3 decoders. As depicted in Fig. 2, the training of VIRT includes two phases, task-unspecified pre-training and task-specified training. In the pre-training phase, a vast collection of robotic manipulation videos, along with corresponding proprioception data, is employed to pre-train a task-agnostic policy. For a sequence of $T$ timesteps, the observations at the first and last timestamp, denoted as $o_1$ and $o_T$, are input to the policy for predicting the manipulation actions of the entire sequence $\{a_1, a_2, \ldots, a_T\}$. After this phase, the pre-trained policy learns rich general manipulation knowledge, and the parameters of the encoders and decoders in the pre-trained policy are utilized to initialize the policy weights in subsequent task-specific training.

During the task-specific training phase, we adopt the action chunking protocol Zhao et al. (2023) to control the robot. Specifically, at the timestamp $t$, a step of image observation $I_t$ and $k$ steps of historical prioperception measurements $\{p_{t-k+1}, p_{t-k+2}, \ldots, p_t\}$ (*e.g.*, the rotation angles, angular velocities, and torques of robot joints) are input to $\pi$ to regress the subsequent $n$ steps of actions $\{a_{t+1}, a_{t+2}, \ldots, a_{t+n}\}$ (named as an action chunk), corresponding Laplacian uncertainty values Li et al. (2022) $\{\sigma_{t+1}, \sigma_{t+2}, \ldots, \sigma_{t+n}\}$, and current status $s$. Among the attributes, $\{a_{t+i}\}_{i=1}^n$

Figure 2: The overall pipeline of the proposed techniques, including robotic imagery pre-training and robotic gaze. In robotic imagery pre-training, the VIRT model is pre-trained with numerous manipulation data. Then, the weight of the pre-trained encoders and decoders is employed to initialize the model in the task-specified training phase, and the robotic gaze is applied to this phase.

are the action chunk to perform. After completing the execution of the current action chunk, the policy $\pi$ updates its observations and predicts the next $n$ actions. $\{\sigma_{t+i}\}_{i=1}^{n}$ represent the prediction uncertainties of $\{a_{t+i}\}_{i=1}^{n}$ and are learned by minimizing the following loss:

$$L_a = \frac{1}{n}\sum_{i=1}^{n}(\frac{\sqrt{2}|a_{t+i} - \hat{a}_{t+i}|}{\sigma_{t+i}} + \log \sigma_{t+i}), \tag{1}$$

where $\{\hat{a}_{t+i}\}_{i=1}^{n}$ denote the action labels collected by human demonstration. According to the formulation in Eq. 1, we can observe that the learned uncertainties $\{\sigma_{t+i}\}_{i=1}^{n}$ exhibit higher values for more ambiguous action segments. Consequently, larger $\{\sigma_{t+i}\}_{i=1}^{n}$ values result in a smaller penalization for the discrepancies between predicted actions $\{a_{t+i}\}_{i=1}^{n}$ and demonstrated actions $\{\hat{a}_{t+i}\}_{i=1}^{n}$. This property of $\{\sigma_{t+i}\}_{i=1}^{n}$ enables $\pi$ to concentrate on more deterministic action segments, which are pivotal for successful manipulation. In addition, the predicted attribute $s$ is a one-hot vector indicating the current stage of manipulation in which the policy $\pi$ is engaged. Specifically, for a long-term manipulation task, the trajectory is manually segmented into multiple stages, with each stage focusing on a specific object for $\pi$. Once the attribute $s$ signifies the completion of the current stage, the detector, as depicted in Fig. 2, directs $\pi$ to focus on the object corresponding to the next stage. Further details on this strategy are provided in Section 3.4.

## 3.3 ROBOTIC IMAGERY PRE-TRAINING

The RIP paradigm enables large-scale robotic manipulation pre-training without the demand for task description annotations. In this paradigm, we provide the first and last timestamps of observations in a manipulation segment to $\pi$ and train $\pi$ to imagine the actions within this segment. Formally, as defined before, the observations and demonstration actions are denoted as $\{o_t\}_{t=1}^{T}$ ($o_t$ is the same as $\hat{o}_t$ during training) and $\{\hat{a}_t\}_{t=1}^{T}$ over a data sequence with $T$ timestamps. The observations $o_1$ and $o_T$ serve as the input to $\pi$, where $o_1$ is the current observation and $o_T$ represents the vision instruction. Each $o_t$ consists of two parts, the image component $I_t$ and prioperception information $p_t$. As shown in Fig. 2, the images $I_0$ and $I_T$ are split into uniform patches and then transformed into vision tokens by Transformer encoders. Similarly, $p_o$ and $p_t$ are encoded as tokens by a linear projection layer. These tokens, along with $n$ action queries, are input to the Transformer decoders to produce $n$ actions $\{a_t\}_{t=1}^{T}$. The policy weights are updated by computing the $L_1$ loss between the predicted actions $\{a_t\}_{t=1}^{T}$ and demonstration actions $\{\hat{a}_t\}_{t=1}^{T}$.

As RIP eliminates the need for task-specific priors or manual annotations, it is widely applicable to diverse robotic manipulation data. In this work, we pre-train $\pi$ with Droid Khazatsky et al. (2024), which is a large-scale in-the-wild robotic manipulation dataset with 76k trajectories. Droid is selected due to its extensive diversity and inherent complexity, as many tasks within Droid are

difficult to describe linguistically and lack task description annotations. Although previous works have attempted to pre-train policies using this dataset, they find that incorporating Droid into their pre-training paradigms harms the downstream performances Kim et al. (2024b), which implies the great challenge behind this dataset posed by its huge task diversity and ambiguity. By contrast, RIP only utilize vision observations and manipulation trajectories to pre-train policis, avoiding relying on any task-specific prior. This characteristic makes RIP easier to handle the challenges posed by ambiguous task content, thereby yielding highly effective pre-trained policy weights.

### 3.4 ROBOTIC GAZE

Due to the need of considering interations between multiple objects and avoiding collisions, a huge receptive field is critical for policy design in robotic manipulation. Therefore, Transformers are advantageous due to their global receptive field. However, this comes at the cost of reduced sensitivity to image details, which is also crucial for robotic manipulation Carion et al. (2020). In addition, this problem is uneasy to address by simply enlarging the whole image, because doubling the width and height of an image results in $4\times$ of tokens, which means $16\times$ increase in computational complexity. To address this problem, we draw inspiration from the human visual system. It is found that the fovea centralis in a human eye contains cells that differ from those in other regions Willmer & Wright (1945). This difference makes eyes can only perceive a small region very clearly, but it helps maintain a good balance between the processing burden of brain cortex and observation resolution. Interestingly, this biological trait closely resembles the functionality needed in Transformers. Thus, we design the RG algorithm to emulate this gaze mechanism,

Specifically, we employ a lightweight detector like YOLOv10 Wang et al. (2024) to recognize objects of interest. The mature state of 2D object detection allows for the easy acquisition of detectors that meet various application demands. Detectors like YOLOv10 can run at speeds exceeding 100 frames per second on an RTX4090 GPU, resulting in minimal inference latency. However, a detector locates all concerned objects, our aim is for the policy $\pi$ to concentrate on a single crucial region for manipulation at any given moment, akin to the human eye gaze mechanism. To bridge this gap, we divide a manipulation task into multiple stages, each corresponding to a specific object of focus. As described in Section 3.2, the policy $\pi$ is trained to predict a one-hot vector $s$ that classifies the current manipulation stage. With this information, the corresponding region in $I_t$ to concentrate on at the timestamp $t$ is obtained, and this region is the vision instruction. Then, we zoom in this vision instruction to the same resolution as $I_t$. In this way, the vision instruction is concatenated with $I_t$ into a single batch for input into the encoders, enhancing inference efficiency and enabling $\pi$ to perceive object details more clearly. Importantly, directing $\pi$ to focus on a single object at a time does not preclude the robot from manipulating multiple objects simultaneously. Similar to human behavior, where the eyes can only gaze at one object while handling multiple, the robot can perform concurrent operations on several objects.

In addition, as discussed in Sec. 1, this RG strategy also imitates the gaze anchoring hypothesis in cognitive science, as the vision instruction produced by RG prompts $\pi$ about which object to manipulate before the actions start. It helps scheduling the operation procedures in a task. Benefiting from this effect, RG significantly mitigates the optimization challenges inherent in long-term manipulation tasks by offering $\pi$ continuous feedback on task progress. Our experimental results suggest that a naive baseline policy without RG struggles to learn to perform challenging long-term tasks, while the policy with RG achieves high success rates.

### 3.5 POLICY DETAILS

In task-unspecified pre-training, the encoders of the VIRT model are initialized using the weights of DINOv2 encoders Oquab et al. (2024), and the other components are initialized randomly. The optimization objective of this phase is solely the $L_1$ loss between the predicted action trajectories $\{a_t\}_{t=1}^T$ and corresponding labels $\{\hat{a}_t\}_{t=1}^T$ obtained through teleoperation. During task-specified training, the encoders and decoders of VIRT are initialized from the RIP pre-trained VIRT policy, with the remaining modules initialized randomly, and the detector is frozen. The loss for status prediction is a standard cross-entropy loss, denoted as $L_s$, contributing to a total loss function as $L = L_a + 100L_s$. The model parameters are updated using the AdamW optimizer Loshchilov (2017) in both training phases. The learning rate is set to $1e-5$. No data augmentation is employed.

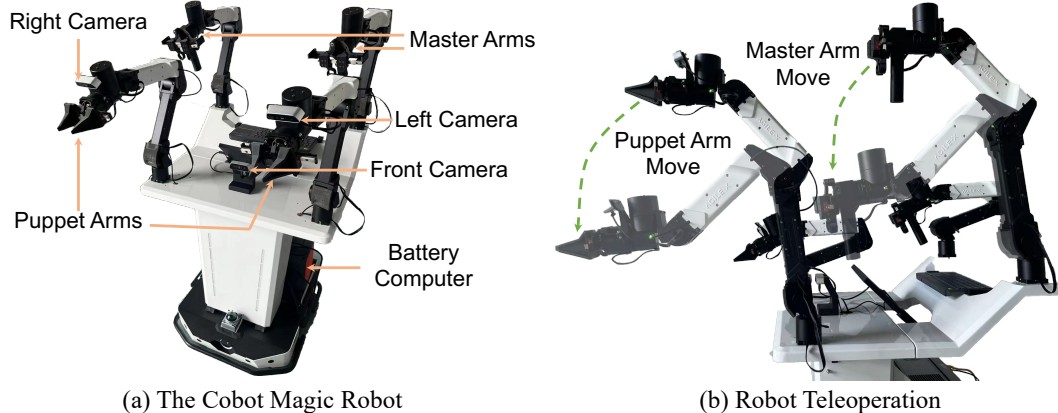

(a) The Cobot Magic Robot

(b) Robot Teleoperation

Figure 3: Illustrations of the Cobot Magic robot and how it is teleoperated. The robot has two master arms and two puppet arms. When collecting demonstration data, the puppet arms imitate the actions of the master arms. In inference, the policy directly control the movements of the puppet arms.

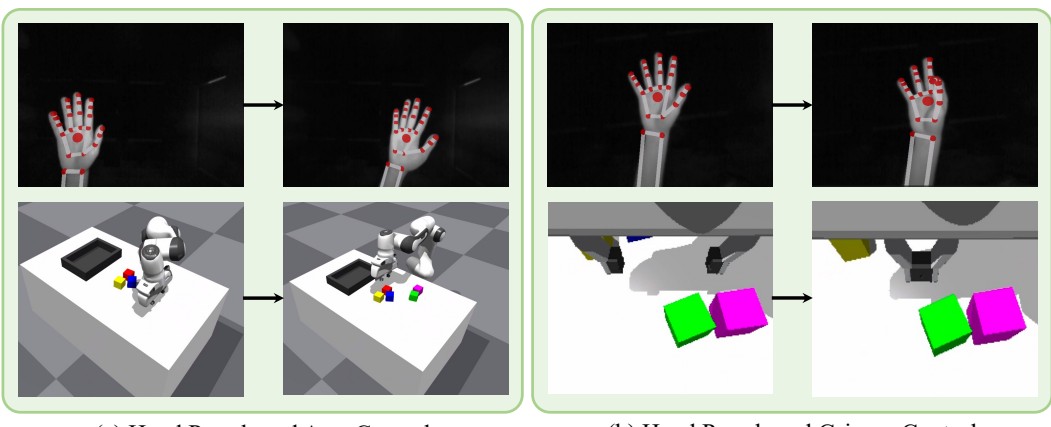

(a) Hand Pose based Arm Control

(b) Hand Pose based Gripper Control

Figure 4: Illustrations of how we teleoperate the robot in Isaac Gym. Specifically, a real-time hand pose acquisition system is built to map the human hand pose to the joint rotations of the robot. We utilze the orientation and translation of the palm to control the end of the robot arm and employ the distance between the thumb and index finger to determine the opening or closing of the gripper.

## 4 EXPERIMENTS

This section seeks to address the following questions: (1) Does RIP pre-training substantially enhance task success rates? (2) Does RG effectively improve scheduling and detail perception capabilities? (3) Can VIRT achieve greater precision in manipulations compared to previous policies?

### 4.1 EXPERIMENT PLATFORMS

**Real-robot platform**. We conduct experiments using the Cobot Magic robot Agilex (2024) to verify the effectiveness of VIRT. As shown in Fig. 3 (a), the robot is integrated with four robotic arms, *i.e.*, two master arms and two puppet arms. During the process of collecting the demonstration data $\mathcal{D}$, we manually control the master arms, and the puppet arms imitate the actions of the master arms in real time. After the policy $\pi$ is trained on $\mathcal{D}$, it directly controls the puppet arms during the inference stage. Three cameras are installed on the robot, which are the right camera, front camera, and left camera, respectively. These cameras provide different observation views for $\pi$. Besides images captured by cameras, the prioperception information, including the rotation angles, angular velocities, and driving torques of various joints in this robot, is also available.

**Simulation platform**. In this work, we design simulated manipulation tasks based on Isaac Gym Makoviychuk et al. (2021), which supports GPU-based efficient physics simulation. A Franka Panda

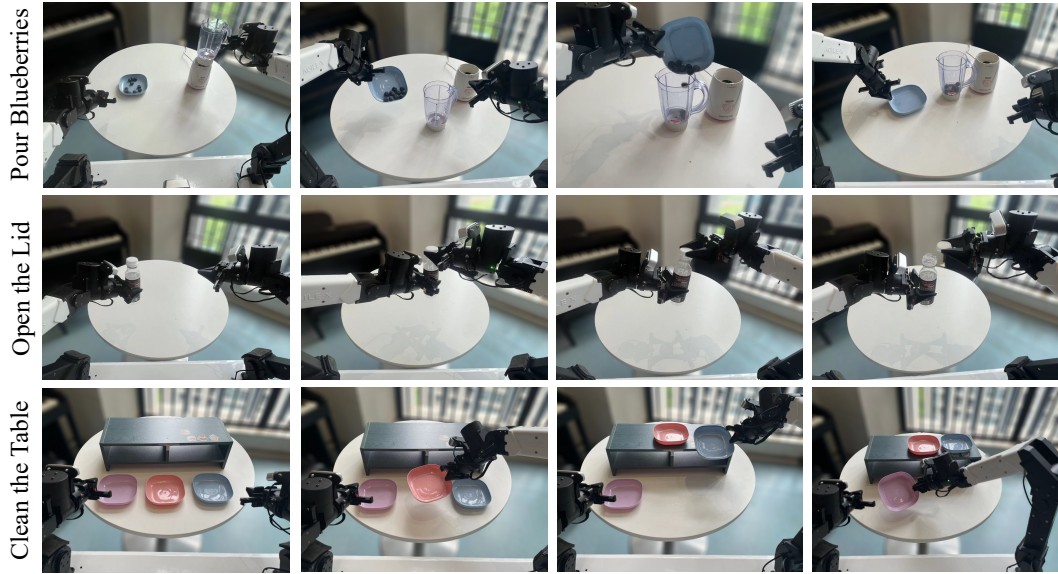

Figure 5: Illustrations of the three designed real-robot tasks, which include Pour Blueberries, Open the Lid, and Clean the Table.

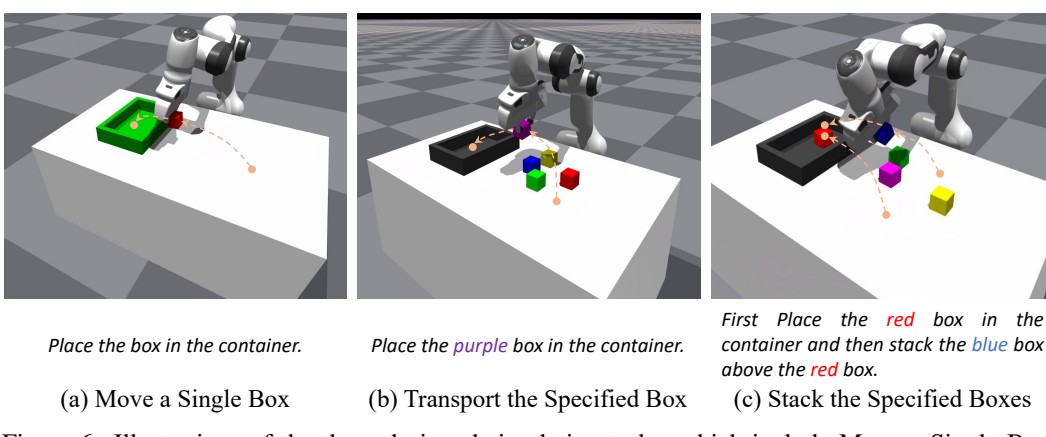

*Place the box in the container.*

*Place the purple box in the container.*

*First Place the red box in the container and then stack the blue box above the red box.*

(a) Move a Single Box      (b) Transport the Specified Box      (c) Stack the Specified Boxes

Figure 6: Illustrations of the three designed simulation tasks, which include Move a Single Box, Transport the Specified Box, and Stack the Specified Boxes.

robotic arm is deployed in each simulation environment to manipulate objects, with four cameras strategically positioned to observe the scene from various angles, including three peripheral views and one hand view. Unlike previous approaches that rely on manually crafted script rules for generating manipulation demonstrations Zhao et al. (2023), we build a real-time hand pose acquisition system to teleoperate the simulated robotic arm, which better mimics the real demonstration data distribution. Specifically, a Leap Motion Controller Ultraleap (2013), which is a binocular infrared camera, is adopted to estimate the hand translation and orientation. This estimation is based on traditional algorithms such as stereo depth inference Blake & Wilson (2011). Then, as shown in Fig. 3 (b), we map the translation and orientation of the hand palm to the robot arm end-effector position using pre-defined rules, and the joint rotation angles of the robot arm are derived based on inverse kinematics Kucuk & Bingul (2006). The opening or closing of the robot gripper is controlled by the distance between the thumb and index finger of the human hand for teleoperation.

## 4.2 TASK DESIGN

To fully evaluate the effectiveness of our proposed techniques, we design three real-robot tasks and three simulated tasks based on the aforementioned experiment platforms.

Table 1: Performance comparison with previous policies in real-robot tasks.

| Policy | PB success ↑ | PB completion ↑ | OL success ↑ | OL completion ↑ | CT success ↑ | CT completion ↑ | Speed ↑ |
|---|---|---|---|---|---|---|---|
| ConvMLP ($n = 1$) | 0.00 | 0.00 | 0.00 | 0.00 | 0.00 | 0.00 | 18.69 |
| ConvMLP ($n = 10$) | 0.00 | 0.02 | 0.00 | 0.06 | 0.00 | 0.05 | 17.54 |
| Diffusion Policy | 0.00 | 0.03 | 0.00 | 0.05 | 0.00 | 0.04 | 27.32 |
| ACT | 0.00 | 0.12 | 0.01 | 0.28 | 0.00 | 0.07 | 43.48 |
| VIRT 🦾 | 0.42 | 0.60 | 0.71 | 0.82 | 0.37 | 0.55 | 39.22 |

Table 2: Performance comparison with previous policies in simulated tasks.

| Policy | MS success ↑ | MS completion ↑ | TS success ↑ | TS completion ↑ | SS success ↑ | SS completion ↑ |
|---|---|---|---|---|---|---|
| ConvMLP ($n = 1$) | 0.00 | 0.00 | 0.00 | 0.00 | 0.00 | 0.00 |
| ConvMLP ($n = 10$) | 0.11 | 0.11 | 0.08 | 0.08 | 0.00 | 0.02 |
| Diffusion Policy | 0.07 | 0.07 | 0.03 | 0.03 | 0.00 | 0.00 |
| ACT | 0.90 | 0.90 | 0.12 | 0.12 | 0.02 | 0.11 |
| VIRT 🦾 | 0.92 | 0.92 | 0.69 | 0.69 | 0.65 | 0.76 |

**Real-robot tasks**. The real-robot tasks are designed for analyzing the capabilities of VIRT from different perspectives. As depicted in Fig. 5, the three tasks are named as Pour Blueberries, Open the Lid, and Clean the Table, respectively. We collect 100 demonstrations of data for every task.

In the Pour Blueberries task, the robot needs to remove the juicer cup from the juicer and place it on the table. Then, the robot picks up the plate containing blueberries and pours all blueberries into the juicer cup. Finally, the plate is returned back to the table. This task is to measure the long-term multi-object manipulation ability of policies, and diverse kinds of actions are required in this task.

For the Open the Lid task, the robot uses a robotic hand to hold a bottle with a tightly screwed lid. The another hand first needs to grasp the lid. After a series of twists, the robot gradually unscrews and removes the lid from the bottle. This task tests the dexterous manipulation capability of policies.

Within the Clean the Table task, three plates of different colors and a small cabinet are positioned on a table. The robot is required to move the plates onto the cabinet in a color order that is randomly specified during test. The orders are different among various trials. This task is to verify the instruction following performance of policies.

**Simulation tasks**. The three real-robot tasks are quite challenging and some policies could get zero success rates. To analyze these policies more sufficiently, we devise three simulation tasks of varying difficulty levels, namely Move a Single Box, Transport the Specified Box, and Stack the Specified Boxes. The three tasks are visualized in Fig. 6. We collect 50 demonstrations for the first task due to its lower difficulty and 100 demonstrations for each of the other two tasks.

In Move a Single Box, the robot needs to transports the sole box on a table in a container. For Transport the Specified Box, five different colors of boxes are randomly located on a table, and the robot should move the box described by a random instruction to the container. Different from Transport the Specified Box, two boxes are specified in Stack the Specified Boxes, and the robot is expected to move the first box in the container and then stack the second box on the first box. Therefore, both Transport the Specified Box and Stack the Specified Boxes test the instruction following ability, and Stack the Specified Boxes demands better long-term operation and precise manipulation capabilities.

**Evaluation metrics**. The success rate of whether completing a task is the primary metric adopted by previous works Chi et al. (2023). Nevertheless, for the tasks requiring multiple steps of operations, this metric does not reflect the completion ratio when a policy does not fullfill a task. To bridge this gap, we design a new metric named the completion score. For a given task comprising $k$ steps, the policy earns a score of $i/k$ if it completes up to the $i_{\text{th}}$ step. Four of the designed tasks contain multi-step manipulation, and they are Pour Blueberries, Open the Lid, Clean the Table, and Stack the Specified Boxes. Refer to Appendix A.1 for details of how the steps are defined. In the following experiments, we test each policy on every task for 100 times and report the average results.

## 4.3 COMPARISON WITH PREVIOUS POLICIES

In this section, we compare VIRT with existing mainstream policies, including ConvMLP Zhang et al. (2018), Diffusion Policy Chi et al. (2023), and ACT Zhao et al. (2023). Among them, ConvMLP is the most commonly adopted baseline, which first extracts image feature using convolutional neural network (CNN) and then regresses actions based on the extracted feature. However,

Table 3: Ablation study on the proposed techniques.

| RG | enlarge | RIP | uncern | TS success ↑ | TS completion ↑ | SS success ↑ | SS completion ↑ | OL success ↑ | OL completition ↑ |
|----|---------|-----|--------|--------------|-----------------|--------------|-----------------|--------------|--------------------|
|    |         |     |        | 0.11 | 0.11 | 0.05 | 0.14 | 0.00 | 0.31 |
| ✓  |         |     |        | 0.32 | 0.32 | 0.24 | 0.50 | 0.26 | 0.48 |
| ✓  | ✓       |     |        | 0.47 | 0.47 | 0.41 | 0.61 | 0.39 | 0.55 |
| ✓  | ✓       | ✓   |        | 0.64 | 0.64 | 0.53 | 0.72 | 0.45 | 0.65 |
| ✓  | ✓       | ✓   | ✓      | 0.69 | 0.69 | 0.65 | 0.75 | 0.71 | 0.82 |

our experiments suggest that the manipulation performance of ConvMLP is poor. Our analysis reveals that this is because the popular implementation of ConvMLP only supports predicting an action chunk with the size of 1 ($n = 1$), resulting in inconsistent action sequences between different predictions. To address this problem, we improve its implementation to support $n = 10$. In the following, we report the results of ConvMLP with both $n = 1$ and $n = 10$. Different from ConvMLP, Diffusion policy decodes the action chunk through iterative denoising. ACT consists of a CNN backbone, encoders, and decoders. Its basic architecture is similar to VIRT and can be treated as a baseline. For the tasks needing to follow manipulation orders randomly generated in test, we encode the text describing the manipulation order as tokens using the CLIP text encoder Radford et al. (2021) and add the encoded tokens to the feature of the compared policies.

We compare VIRT with these policies using the designed real-robot tasks (PB: Pour Blueberries, OL: Open the Lid, CT: Clean the Table) and simulation tasks (MS: Move a Single Box, TS: Transport the Specified Box, SS: Stack the Specified Boxes) and report the results in Table 1 and Table 2. The inference speeds of these policies, which are test using a RTX4090 GPU, are also presented. We can observe that VIRT outperforms the compared polices by large margins, and the inference speeds are also promising thanks to its efficient implementation. Besides the improvments brought by our proposed techniques, we further investigate potential performance constraints in the compared policies. For ConvMLP and Diffusion Policy, a critical problem is that their network heads for predicting actions are implemented based on fully connected layers, the computation burdens of which are quite heavy. To alleviate this problem, ConvMLP and Diffusion Policy have to compress the image feature into smaller embedding. For example, Diffusion Policy compresses image feature as keypoint embedding using the SpatialSoftmax module Finn et al. (2016), and this compression causes the policy cannot receive sufficient observation information for manipulation. Differently, the problem of ACT is its weakness in handling the ambiguity in manipulation. For example, according to the results in Table 2, ACT achieves similar performance with VIRT in the Move a Single Box task, where there is only a sole box in the environment. However, when we test ACT in the Transport the Specified Box task, its results become much poorer due to the increasing ambiguity.

### 4.4 Ablation Study

We perform an ablation study of the proposed techniques using the Transport the Specified Box (TS), Stack the Specified Boxes (SS), and Open the Lid (OL) tasks. In these tasks, TS and SS test the instruction following and multi-step operation capabilities of policies, and OL mainly verifies the dexterous manipulation precision. The results are presented in Table 3. Our findings indicate that the baseline policy, when not incorporating the proposed techniques, achieves similar performance to the results of ACT reported in Table 1 and Table 2. Then, by incorporating the developed techniques, the manipulation performances of VIRT are boosted significantly through addressing the unclear observation, and deficient feature, and trajectory ambiguity problems.

## 5 Conclusion

In this work, we have demonstrated vision observations are more suitable for serving as manipulation instructions than text descriptions. Afterwards, inspired from the human biological mechanisms, we have proposed the RIP and RG strategies for applying vision instruction to the task-unspecified pre-training and task-specified training in robotic manipulation learing. Based on these two strategies, a fully Transformer-based policy VIRT has been developed. To evaluate the effectiveness of VIRT, we have designed three real-robot tasks using the Cobot Magic robot and three simulated tasks based on Isaac Gym. These six tasks test the manipulation capabilities of policies from different perspectives. Our sufficient experiments suggest that VIRT have surpassed the performances of compared popular policies by large margins and all the developed techniques are effective.

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

## A   Appendix

### A.1   Step Definitions in Tasks

In this work, we define a new metric, the completion score, to reflect the task completion ratio when a policy fails to fulfill the whole task. In our six devised tasks, four of them include multiple steps of operations, and the tasks are Pour Blueberries, Open the Lid, Clean the Table, and Stack the Specified Boxes. All these tasks include three operation steps, and we explain how the three steps are defined in these tasks as follows:

**Pour Blueberries**: The first step is taking the juicer cup off the juicer and placing the juicer on the table successfully. The second step is picking up the plate containing blueberries. The third step is pouring blueberries into the cup successfully.

**Open the Lid**: The first step is that the robotic hand for screwing the bottle lid grasps the lid correctly. The second step is the robotic hand screws the lid in the correct motion. The third step is the lid takes the lid from the bottle successfully.

**Clean the Table**: As there are three plates in various colors on the table, the $i_{\text{th}}$ step is moving the $i_{\text{th}}$ specified plate on the cabinet successfully.

### A.2   Manipulation Videos

We have provided videos recording the process of the VIRT policy controls robots to complete the six designed tasks, and the video files can be found in Supplementary Material.

### A.3   Code

We provide the source code as the file "VIRT_code.zip" in Supplementary Material. The code, data, and pre-trained policy weight will be publicly available soon.

