# OpenReview forum: "VIRT: Vision Instructed Transformer for Robotic Manipulation"
_ICLR.cc/2025/Conference — ICLR 2025 Conference Withdrawn Submission_

### Official Review · Reviewer_FbPm · 2024-10-26

**Soundness:** 2
**Presentation:** 2
**Contribution:** 2
**Rating:** 3
**Confidence:** 5

**Summary:**

This paper studies the problem of visuomotor policy learning for manipulation tasks.
The method has two phase. 1) Robotic Imagery Pre-training that trains an image-conditioned policy for many tasks and 2) robotic gaze strategy: use an object detector to detect the objects of interest in each stage of the task and use that as one of the inputs to the policy.

**Strengths:**

It is able to demonstrate positive pretraining results with the DROID dataset, which has not been demonstrated before.

The paper demonstrates results in both simulated and real-world environments.

**Weaknesses:**

Despite the paper naming the method with new names e.g., Robotic Imagery Pre-training or robotic gaze strategy. I don't see this paper introducing significantly new concepts or method.  Imagery Pre-training could be considered just as a standard image-conditioned policy.  Robotic gaze strategy can be considered as conditional policy with fixed stages, where the condition is the stage and object detection result.


The paper says "RIP eliminates the need for task-specific priors or manual annotations" however, for robot data, the most expensive part to obtain is the robot action, annotation on task is relative easy. If the pretraining still require robot action (teleoperated data), I'm not sure how much more scalable it is compare to existing approaches.

Dividing tasks into stages, where each stage only has one object of interest, is a strong assumption for unstructured manipulation tasks -- How do we define the stage for any manipulation task consistently? What if the demonstrator does the tasks in different orders or stages? What if there are multiple objects of interest?

The paper writing is a bit confusing -- after reading the abstract and introduction, I don't have a good idea what is the " imagery paradigm" the vision instruction for tasks. Or the Transformer-based policy network architecture. I feel the paper could reduce the motivation and connections to cognitive science and instead be more direct about the technical contribution to improve clarity.

It is surprising to see the task performance for prior works (both ACT and Diffusion Policy) are so low, especially for simulation tasks, since it does not seem particularly challenging. Also, the paper does not provide an evaluation of the existing benchmarks, so it is not clear whether the implementation of baselines is correct.

**Questions:**

Why changing the network between pretraining and finetuning? Why not directly train the finetuning network with pretraining data? The pretraining data also contains robot action meaning it can be used to directly train the final network.

Why not provide some evaluation on existing robot manipulation benchmarks?

---

### Official Review · Reviewer_2NE6 · 2024-10-28

**Soundness:** 1
**Presentation:** 2
**Contribution:** 1
**Rating:** 3
**Confidence:** 4

**Summary:**

This paper introduces VIRT (Vision Instructed Robotic Transformer), a Transformer-based model designed to improve robotic manipulation by using vision-based instructions rather than text. Drawing inspiration from cognitive science, the authors propose two key methods: Robotic Imagery Pre-training (RIP) and Robotic Gaze (RG). RIP enables large-scale pre-training by allowing the model to "imagine" action sequences from initial and final visual states, while RG emulates human eye-gaze to focus on task-critical objects. VIRT was tested on complex real-world and simulated tasks, such as manipulating multiple objects and performing dexterous actions, where it significantly outperformed text-instructed models, achieving high success rates. The study concludes that vision-based guidance can improve robotic performance in understanding and executing manipulation tasks with precision.

**Strengths:**

1. RIP allows large-scale training without extensive labeled data.
2. Robotic Gaze Focuses attention on key objects, boosting success in complex tasks.

**Weaknesses:**

1. The contributions of this paper could be articulated with greater clarity and supported by a more comprehensive evaluation. The paper presents two key contributions: **Robotic Imagery Pre-training** and **Robotic Gaze**. However, while each contribution has potential, the overall impact could be enhanced with further refinement and empirical comparison:
    1. Robotic Imagery Pre-training: The paper suggests that prior work in pretraining typically relies on text annotations, which is presented as a limitation. However, integrating language as part of the design is often a deliberate choice, considering its utility for task specification in manipulation tasks, where language enables easy and flexible task specification. Similar to the proposed approach, using a goal image as a conditioning factor is well-established, especially in navigation tasks, as seen in works like ViNT[1] and NoMaD[2]. Given this context, I would suggest clarifying the novelty of using goal images for pretraining in this work. Furthermore, the evaluation could benefit from including comparisons with alternative pretraining methods to strengthen the case. The original DROID paper has demonstrated the benefits of pertaining. It would also be valuable to include a rationale for the choice of goal image for pretraining task specification. In addition, it is unclear how such pretraining setting would benefit downstream tasks.
    2.  Robotic Gaze: The robotic gaze component is intriguing and has promising potential. However, the current presentation lacks clarity, particularly regarding how the object of interest is determined—whether this is learned by the policy or manually defined. Even if it is manually defined, distinguishing this approach from related methods like visual prompting, which generates affordances based on natural language (e.g., MOKA[3]), could make a stronger case for its contribution. Additionally, a comparison with visual prompting methods might reveal unique advantages or insights that enhance the impact of this work.
2.  A more thorough evaluation could further highlight the contributions. Adding comparisons to established pretraining methods and including common benchmark experiments alongside the customized tasks in the paper would provide a well-rounded assessment.

To strengthen the paper, I would recommend focusing on a single primary contribution and conducting a more in-depth evaluation and analysis. For example, the paper could either concentrate on demonstrating how goal-image-conditioned pretraining facilitates better task understanding and generalization or explore how robotic gaze enables the policy to acquire complex skills.

[1] ViNT: A Foundation Model for Visual Navigation

[2] NoMaD: Goal Masked Diffusion Policies for Navigation and Exploration

[3] MOKA: Open-World Robotic Manipulation through Mark-Based Visual Prompting


Minor:
1. Please consider using $\citep$ instead of $\cite$.
2. The introductory quotes, while intriguing, could benefit from a clearer connection to the core contributions and themes of the paper. I do not see how it is connected with the main contribution or argument.

**Questions:**

see weaknesses.

---

### Official Review · Reviewer_ZwkC · 2024-11-03

**Soundness:** 2
**Presentation:** 3
**Contribution:** 2
**Rating:** 3
**Confidence:** 3

**Summary:**

The paper studies the problem of learning visuo-motor control policy through behavior cloning. The proposed framework uses goal images as visual instructions to specify the tasks. The model is first pre-trained with large-scale dataset and fine-tuned with in-context robot manipulation data collected through tele-operation. The method is evaluated on three real world tasks and three simulation tasks.

**Strengths:**

The problem is well defined, and the proposed method is straightforward and standard.

The performance of the method outperforms all baselines from the evaluation perspective.

The paper is well structured and easily read.

**Weaknesses:**

Given that many prior works [1][2][3][4] utilize similar model architectures/goal representations for learning visuo-motor policies, the technical contribution of the work is diminished.

The comparison with baselines insufficient. More relevant approaches (e.g., [4][5]) should be compared and discussed.

For real world experiments, the success rates for ACT and diffusion policy are extremely low, could authors give more explanations and analysis on this?

[1] Multimodal Diffusion Transformer: Learning Versatile Behavior from Multimodal Goals, RSS’24;

[2] Goal Conditioned Imitation Learning using Score-based Diffusion Policies, RSS’23;

[3] MimicPlay: Long-Horizon Imitation Learning by Watching Human Play, CoRL’23;

[4] ALOHA Unleashed: A Simple Recipe for Robot Dexterity, CoRL’24.

[5] 3D Diffusion Policy: Generalizable Visuomotor Policy Learning via Simple 3D Representations, RSS’24

**Questions:**

What are the failure modes of the proposed method and baselines?

In the Supplementary Material, there is only one video for each real-world task, what are the reset ranges of objects? Providing additional details would help readers better understand the generalizability of the method.

---

### Official Review · Reviewer_8q6R · 2024-11-03

**Soundness:** 3
**Presentation:** 3
**Contribution:** 3
**Rating:** 5
**Confidence:** 3

**Summary:**

The paper introduces the Vision Instructed Transformer (VIRT), a novel model for robotic manipulation that leverages vision-based instructions instead of natural language. It aims to overcome the limitations of natural language-based instructions by introducing two key components: (1) Robotic Imagery Pre-training (RIP), a pre-training paradigm using visual-only data to improve scalability and avoid expensive image-text alignment, and (2) Robotic Gaze (RG), which emulates human eye gaze to focus on the object of manipulation. Together, these mechanisms enable VIRT to significantly improve performance in complex manipulation tasks, as evidenced by results from real-robot and simulated environments.

**Strengths:**

1- The paper proposes a vision-centric approach to robot manipulation, addressing the scalability issue of text-instruction methods. The focus on vision instructions could be impactful, offering a more practical path for real-time robotic applications.

2- The RIP module is an interesting concept inspired by cognitive science's "imagination" mechanism, allowing the model to generalize manipulation skills without relying on textual annotations. This pre-training approach is valuable in that it removes dependencies on expensive, labeled data.

3- The RG module is designed to enhance focus on target objects by cropping and enlarging specific areas in the visual input, similar to hard attention mechanisms. The approach effectively balances computational efficiency with resolution demands, which is a critical factor in robotic manipulation.

4- The experimental results are promising. The model outperforms baseline approaches across various challenging tasks, demonstrating the effectiveness of RIP and RG in real-world robotic manipulation.

**Weaknesses:**

1- While the study provides insights, the analysis could be more comprehensive. For example, the role of the uncertainty score is somewhat underexplored. Further exploration on how the score manages discrepancies between predicted and actual actions, especially in uncertain segments, would add value. Additionally, it would be insightful to see the impact of only using RIP without RG fine-tuning.

2- Figures 1 and 2 are somewhat inconsistent with the textual description. For instance, Figure 1 only shows image inputs, though VIRT also uses proprioceptive data. Additionally, the “Query Chunk” element in Figure 2 lacks clarity, and it would help if the paper provided more detail on this element's purpose and its role in both RIP and RG modules.

3- Despite the advantages of a vision-based approach, the method relies heavily on object detectors, which might limit its adaptability to novel or unseen objects. Handling scenes with multiple, visually similar objects could be challenging and would benefit from further clarification.

4- The experimental section could benefit from clearer task definitions. The number of task segments, initial observations, target vision instructions, and object crop regions should be explicitly shown for each task. Additionally, the segmentation of tasks into stages should be thoroughly explained in the methodology section for clarity.

**Questions:**

1- Could the authors clarify the purpose of the "Query Chunk" and "action queries"? Are they initialized randomly? Additionally, are additional heads added during the RG fine-tuning phase?

2- How are long-horizon tasks segmented into segments, as mentioned in Section 3.2? Could you elaborate on the criteria or methods used?

3- What is the performance impact of omitting RG fine-tuning? Including this analysis in the ablation study could provide valuable insights.

4- How does the proposed method compare with text-image instruction methods? A discussion or comparison of the advantages and limitations would be beneficial.

---

### Note · Authors · 2024-11-18

I have read and agree with the venue's withdrawal policy on behalf of myself and my co-authors.